# Epidemiology, Transmission Mode, and Pathogenesis of *Chlamydia pecorum* Infection in Koalas (*Phascolarctos cinereus*): An Overview

**DOI:** 10.3390/ani14182686

**Published:** 2024-09-15

**Authors:** Mohammad Enamul Hoque Kayesh, Md Abul Hashem, Kyoko Tsukiyama-Kohara

**Affiliations:** 1Department of Microbiology and Public Health, Faculty of Animal Science and Veterinary Medicine, Patuakhali Science and Technology University, Barishal 8210, Bangladesh; 2Department of Cell and Developmental Biology, Feinberg School of Medicine, Northwestern University, Chicago, IL 60611, USA; md.hashem@northwestern.edu; 3Transboundary Animal Diseases Centre, Joint Faculty of Veterinary Medicine, Kagoshima University, Kagoshima 890-0065, Japan

**Keywords:** chlamydia, koala, *Chlamydia pecorum*, chlamydiosis

## Abstract

**Simple Summary:**

Chlamydial infections are a major threat to the health of koalas (*Phascolarctos cinereus*), an iconic Australian marsupial. Among the different chlamydial species, *Chlamydia pecorum* (*C. pecorum*) is the major pathogen infecting koalas, affecting their health and long-term survival, both in the wild and in captivity. Therefore, a deeper understanding of chlamydial infections, including their epidemiology, transmission mode, pathogenesis, host immune response, control, and prevention, is critical for the management of chlamydial infections in koalas. Herein, we discuss the current literature on *C. pecorum* infection in koalas, including the epidemiology, transmission, pathogenesis, immune response, and control strategies for chlamydial infection, with the aim of improving koala health and achieving effective conservation strategies.

**Abstract:**

Chlamydial infections pose a significant threat to koala populations. *Chlamydia pecorum* (*C. pecorum*) remains the major chlamydial species affecting koala health, both in the wild and in captivity, and chlamydial infections are considered important factors affecting the long-term survival of koalas. A clear understanding of chlamydial infections, including the epidemiology, transmission mode, pathogenesis, immune response, control, and prevention thereof, is essential for improving the management of chlamydial infections in koalas. In this study, we discuss the important advances made in our understanding of *C. pecorum* infection in koalas, focusing on the epidemiology of chlamydial infections, and the transmission, pathogenesis, immune response, and control strategies for chlamydial infection, with the aim of improving koala health and achieving effective conservation strategies.

## 1. Introduction

Koalas (*Phascolarctos cinereus*) are an endangered marsupial species threatened by extinction from various factors, including obligate intracellular bacterial infections and chlamydiosis [1,2]. Chlamydiosis is a well-documented and important disease in koalas that is characterized by ocular, urinary, and reproductive lesions [2,3]. Chlamydial ocular infections can lead to blindness, and genital tract infections can cause infertility, among other serious clinical manifestations [2]. *Chlamydia pecorum* (*C. pecorum*) is globally known as ‘koala chlamydia’ [4]. *C. pecorum* gastrointestinal tract (GIT) infection has been shown to be associated with urogenital tract infections in koalas [5]. It has also been reported that *C. pecorum* is present in the reproductive tracts of both male and female koalas [6], and that the male reproductive tract can act as a reservoir for persistent chlamydia infections in koalas [3]. In our previous study, we observed a 21.74% prevalence of *C. pecorum* in captive koalas across Japanese zoos, with female koalas having a higher prevalence rate (24.24%) than male koalas (15.38%) [7]. In Japanese zoos, a higher *C. pecorum* load was observed in adult koalas than in other groups [7]. However, no direct association between koala health and *C. pecorum* load has been observed in captive koalas in Japanese zoos [7].

Important chlamydial species, including *C. pecorum* and *Chlamydia pneumoniae* (*C. pneumoniae*) can cause serious infections in koalas [2,8]. The majority of previous research has focused on *C. pecorum* because it is recognized as a more prevalent and pathogenic species in koalas than *C. pneumoniae* [3,9,10,11,12,13]. Chlamydiosis plays a significant role in koala population decline [2,14], and *C. pecorum* infection is believed to be one of the major pathogens responsible for chlamydiosis in koalas, affecting their long-term survival [15,16]. Robbins et al. reported chlamydial genotype-based variations in pathogenicity, where genotypes belonging to both multi-locus sequence-typing sequence type (ST) 69 and ompA genotype F were linked to disease progression, whereas ST 281 was linked to the absence of disease [17].

*C. pecorum* is considered the key chlamydial pathogen infecting koalas, and complex host–pathogen interactions exist. Therefore, a clear understanding of the prevalence, transmission, pathogenesis, and preventive tools, such as the development of vaccines against *C. pecorum*, is essential for improving koala health and conservation, both in the wild and in captivity [16]. Therefore, in this review, we focus on the epidemiology of *C. pecorum* infection, the transmission mode, pathogenesis, and effects on the host, as well as control and preventive strategies for the improved management of the koala population.

## 2. Epidemiology of Chlamydiosis

A thorough understanding of chlamydial epidemiology and disease dynamics in koalas is critical for improved disease management [18]. However, there is a scarcity of comprehensive and longitudinal population studies that provide a proper understanding of chlamydial infections [18,19]. *C. pecorum* is a significant pathogen in both domestic livestock and wildlife, including koalas [20]. *C. pecorum*-specific multi-locus sequence analysis revealed that Australian koala isolates formed a distinct clade, with limited clustering with *C. pecorum* isolates from Australian sheep [20].

Many studies have reported variations in the prevalence of *C. pecorum* infections in koalas. *C. pecorum* infections are endemic to free-ranging koalas in South East Queensland, where 30% of the surveyed animals exhibited clinical chlamydial disease [13,16]. In contrast, Burach et al. reported 9% (2/23) *C. pecorum* prevalence in Queensland koalas [9]. In addition, chlamydial genotypes were found to differ in two geographically isolated South East Queensland koala populations [17]. A higher prevalence of genotype B, which is considered less pathogenic, was reported in Victorian koalas [12]. However, *C. pecorum* genotype B has not been reported in northern (Queensland and New South Wales) koalas [12]. The prevalence of *C. pecorum* was also much higher in Victorian koala populations on Raymond Island (43/104; 41.3%) than in the Mount Eccles National Park koala populations (30/120; 25%) [11]. However, *C. pecorum* was not detected in French Island koala populations (*n* = 63), and *C. pneumoniae* was not observed in three Victorian koala populations, suggesting geographical differences in chlamydia infections [11]. Moreover, *C. pecorum* was detected only in urogenital swabs and not in ocular swabs, and no ocular disease was observed [11], suggesting that ocular and urogenital swab results could indicate differential pathogenesis in this population.

In a study by Fabijan et al. on South Australian wild koala populations, a 46.7% *C. pecorum* prevalence was observed in mainland Mount Lofty Ranges koalas (*n* = 75), whereas Kangaroo Island koalas (*n*  =  170) were free of *C. pecorum* infection [15], indicating geographical variation in *C. pecorum* prevalence. A summary of the epidemiology of *C. pecorum* infection in Australian koala populations is shown in Figure 1.

## 3. Transmission Mode of *Chlamydia pecorum* in Koalas

*Chlamydia* poses a significant health concern to various species, particularly koalas, and has become pervasive in koala communities, affecting both captive and wild populations. *Chlamydia* can spread systemically in the inner organs of the koala [9], as shown in Figure 2; understanding the transmission dynamics of this bacterial infection in koalas is crucial for implementing effective conservation strategies. *Chlamydia*, recognized as sexually transmitted pathogens among koalas, is primarily spread through direct contact, including sexual transmission (Figure 2) [2,3,11,21,22]. Another study focusing on the male reproductive tract revealed that it may serve as a reservoir for *C. pecorum*, potentially leading to its sexual transmission to female koalas [3]. However, further studies have expanded our understanding of *C. pecorum* transmission by revealing non-sexual pathways, particularly between mothers (dams) and offspring (joeys), through vertical transmission (Figure 2). Two studies focusing on joeys, one dependent (less than one year old) and the other sexually immature (9 to 13 months old), reported a 27% *C. pecorum* prevalence in joeys (*n* = 15) [16]; (*n* = 11) [23]. The study involving dependent joeys [23], based on captive koalas, suggested dam-to-joey transmission as the primary route (Figure 2), although handling by the same animal handler was not entirely ruled out. Conversely, a sexually immature joey study [16] conducted in a monitored wild koala population deemed routes other than dam-to-joey transmission unlikely. These findings underscore the importance of considering both sexual and non-sexual transmission routes when managing *Chlamydia* infection in koala populations.

An additional potential pathway for *C. pecorum* transmission is the fecal–oral route. Pap feeding, the intricate process during which the joey consumes maternal cecal material, plays a crucial role in the inoculation of the GIT with essential microbes necessary for efficient digestion. This phenomenon has also been postulated to be a potential vertical transmission route (Figure 2) [25,26]. In a study by Narayan et al. [27], physiological stressors were found to play a significant role in the transmission of *Chlamydia* within koala populations (Figure 2).

The transmission of *C. pecorum* in koalas occurs primarily through direct contact with infected individuals or their bodily fluids. A recent investigation by Casteriano et al. [24] focused on detecting *Chlamydia* DNA in environmental samples obtained from a koala care facility in New South Wales, Australia. This study highlighted contaminated fomites as a plausible source of *C. pecorum* infection in koalas, underlining the importance of considering environmental factors in the transmission dynamics of this pathogen (Figure 2). The *C. pecorum* isolates found in French Island koalas were genetically related to livestock *C. pecorum* genotypes [28], suggesting potential transmission between species. Overall, *C. pecorum* may utilize multiple methods of transmission among koalas; however, sex or contact with infected genital fluids, such as semen or vaginal fluid, remains the primary mode of transmission. Therefore, a proper understanding of the *C. pecorum* transmission mode is critical for limiting *Chlamydia* spread among koala populations.

## 4. Pathogenesis of *C. pecorum* and Its Effects on Koala Health

Infection with *C. pecorum* is common in koalas, and the pathogenesis underlying chlamydiosis is complex. Understanding the intricate stages of *C. pecorum* infection is crucial for managing and reducing its harmful effects on koala health. We present a comprehensive overview of the current understanding, along with recent modifications, in Figure 3.

Chlamydial infections pose a significant threat to the koala. Initially, the bacterium targets mucosal surfaces, including the conjunctiva, urogenital tract, and respiratory tract. However, it is worth noting that infections can often remain subclinical, evading detection for prolonged periods.

Both male and female koalas can be infected with *C. pecorum*, which poses a significant threat to their health. Variations in the disease manifestations of *C. pecorum* infection in different sexes have been reported, where the brown staining of fur around the rump area, colloquially referred to as ‘wet bottom’, was observed in males, and reproductive tract pathology was observed in females [12]. Reproductive tract infections further exacerbate health challenges. In females, these infections can induce inflammation and fibrosis, leading to complications such as cystic enlargement of the ovarian bursae, metritis, salpingitis, pyometra, hydrosalpinx, and vaginitis [26]. Chlamydial infections remain a major contributor to the development of reproductive cysts, resulting in female infertility and euthanasia [29]. The subclinical infection of *C. pecorum* in male koalas is also not uncommon [3]. The penile urethra is the major site of *C. pecorum* infection in male koalas, along with the prostate and bulbourethral glands [3]. Males may experience prostatitis, orchitis, and epididymitis [3,10], which can also lead to infertility [10,26,30,31]. Recent studies by Hulse et al. [22,32] shed light on the additional impact of *C. pecorum* infection. They demonstrated the adverse effects on semen quality in koalas, with disruptions in spermatogenesis and inflammation in wild koala populations. This leads to a decreased sperm concentration, which is accompanied by testicular degeneration and atrophy. These findings underscore the multifaceted nature of *C. pecorum* pathogenesis and its implications for the health of koala populations.

Of particular concern are ocular infections, which can lead to keratoconjunctivitis and, in severe cases, may cause blindness [2,26,33,34]. Urogenital infections present a spectrum of complications, including urethritis, ureteritis, nephritis, and cystitis. A notable sign is the ‘wet bottom,’ which stems from incontinence [2,26]. However, wet bottom, an indication of chronic urinary tract infection, was observed in both *C. pecorum*-infected and non-infected Victorian koalas, suggesting that other causes may underlie this clinical sign [11], requiring further investigation. Patterson et al. also observed that fecundity is inversely proportional to the *Chlamydia* infection rates in koalas [11].

Respiratory tract infections contribute to the manifestation of rhinitis/pneumonia [35] in infected individuals. In a previous study, clinical disease with ocular and urogenital signs was reported in 5 out of 24 *C. pecorum*-infected koalas; however, no clinical signs were observed among seven *C. pneumoniae*-infected koalas, suggesting *C. pecorum* is more pathogenic in koalas than *C. pneumoniae* [21]. The association of *C. pecorum* with pneumonia in juvenile male koalas has been reported [36]. A microscopic examination of the lungs revealed pyogranulomatous bronchopneumonia, characterized by the proliferation of bronchiolar and alveolar epithelia, along with interstitial fibrosis [36].

## 5. Immune Response to Chlamydial Infection

A proper understanding of the host immune response to a particular pathogen is critical for devising prophylactic and therapeutic interventions against diseases or for evaluating the efficacy of any given vaccine [37,38]. The innate immune response is a key component of the host that acts as the first line of immune defense against many viral infections and shapes adaptive immunity [39]. Unfortunately, due to the unavailability or limited availability of koala-specific reagents and other characterization tools, such as antibodies and PCR, the innate immune response characterization in *C. pecorum*-infected koalas remains unclear [40]. However, genetic resources for koalas should enhance the development of characterization tools for studying immune responses in this iconic species [41,42]. Genetic variations, including single nucleotide polymorphisms (SNPs), may influence innate immune responses against invading pathogens and disease outcomes [43,44].

In two northern populations of koalas, Silver et al. observed the association of MHC class I genes with chlamydial disease progression and the SNPs of 17 genes involved in the resolution of *Chlamydia* infection [45], suggesting a role for host genetics in the outcome of chlamydial infection. Other studies have also indicated the association of MHC class II genes with chlamydial disease susceptibility or pathogenesis in koalas [46,47,48]. An association between the expression of the immune cytokine interleukin (IL)-17 has been linked to MOMP vaccination, and animals with a high urogenital chlamydial load exhibit lower IL-17 levels, favoring disease progression [49]. Koala retrovirus (KoRV)-positive koalas had significantly lower levels of IL-17A and interferon (IFN)-γ gene expression compared to KoRV-negative koalas. However, chlamydial infection and the combined effect of KoRV did not affect these populations [50]. Interestingly, it has been previously reported that chlamydial disease in koalas induces significantly higher IL-17A gene expression compared to that in asymptomatically infected animals [51]. In another study, it was reported that IFN-γ was unable to restrict the growth of *C. pecorum* in bovine kidney epithelial cells, while *C. trachomatis* was inhibited, suggesting *C. pecorum* adopts mechanisms to evade immune response as in its natural host [52]. A recent study reported an association between the numbers of CD3, CD4, CD79b, and HLA-DR-positive cells [6], where a higher chlamydial load was associated with a higher inflammatory score, and a low chlamydial load was associated with a lower inflammatory score [6].

## 6. Control and Prevention of Chlamydial Infection

Antibiotics remain the front-line treatment for chlamydial infections in koalas despite their detrimental effects on the gut microbiota, leading to potential dysbiosis and death [33]. Although wild koalas infected with *Chlamydia* are treated with antibiotics in hospitals, a large percentage of infected animals do not survive [29]. Daily injections of chloramphenicol markedly reduced chlamydial shedding, which was undetectable by the end of the 2nd week of treatment [53]. However, a recent study reported the presence of doxycycline and chloramphenicol resistance genes in 9.9% of koalas, which may interfere with positive outcomes during treatment with doxycycline and/or chloramphenicol in chlamydiosis [54]. The development of new treatment strategies is essential to combat antibiotic resistance in koalas. A recent in vitro and in vivo study demonstrated the effectiveness of the serine protease HtrA inhibitor JO146 in treating *C. pecorum* and *C. pneumoniae* infections [55], which could be developed as a novel treatment tool for treating chlamydiosis in koalas.

Vaccines remain invaluable tools for preventing infections, which can be used as the preferred tool to reduce the burden of chlamydial disease in wildlife, including koalas [56]. A proper understanding of the host immune response to chlamydial infections is important for developing effective chlamydial vaccines [57]. Scientists are attempting to develop an effective vaccine to fight widespread chlamydial infection in koalas and its debilitating effects on koala health.

Kollipara et al. demonstrated the induction of antibodies against epitopes in the conserved domains of *C. pecorum* G and H strains of major outer membrane proteins (MOMP) in both chlamydia-free koalas and naturally infected koalas [57], suggesting that further refinement of vaccine candidates may offer widespread cross-protection against a variety of chlamydial infections circulating in wild koala populations. Khan et al. reported the induction of very low levels of *C. pecorum*-specific neutralizing antibodies in naturally infected koalas, whereas a strong induction of *C. pecorum*-specific neutralizing antibodies was observed in recombinant MOMP-vaccinated koalas [58]. In another study, Khan et al. reported that a single dose of a recombinant chlamydial MOMP adjuvanted with either a Tri-Adj or immune stimulating complex (ISC) vaccine could produce strong cellular (IFN-γ and IL-17A) and humoral (recombinant MOMP specific IgG) immune responses in *Chlamydia*-negative koalas [59]. Desclozeaux et al. reported that a single dose of chlamydial MOMP or Polymorphic Membrane Protein (PMP) vaccine could induce both a systemic and mucosal humoral immune response with anti-chlamydial IgG and/or IgA antibodies as well as cell-mediated immune response with increased IFN-γ and IL-17 production [60]. However, only MOMP vaccine candidates exhibited a clearance of infection in all infected koalas [60]. Joeys from vaccinated mothers were also less likely to be infected than those from unvaccinated mothers, suggesting protection from infection through maternal immunization [23].

It has been shown that vaccinated koalas had an increased life-span (a median lifespan of 12.25 years) compared to unvaccinated koalas (a median lifespan of 8.8 years), and no adverse effects were observed in vaccinated koalas [61]. A previous study demonstrated an improvement in ocular disease conditions in free-ranging koalas after therapeutic vaccination with recombinant major outer membrane proteins (rMOMP) of *C. pecorum* genotypes A, F, and G [33,62]. Another recent study reported the effectiveness of the chlamydia vaccine in mild ocular chlamydial disease, which increased the efficacy of antibiotic treatment for cystitis in vaccinated koalas [63]. However, another study reported a contrasting finding for the chlamydia vaccine, where no significant stimulation of the plasma anti-MOMP IgG response was generated in koalas vaccinated with a synthetic peptide vaccine composed of four components of *C. pecorum* MOMP [64]; therefore, further investigation is required.

Chlamydial MOMP vaccine or chlamydial MOMP plus a KoRV recombinant envelope protein (rEnv) vaccine (combined vaccines) were found to be effective against *C. pecorum* infection, and the inhibition of *C. pecorum* became more pronounced over time [1], indicating the efficacy of the vaccine in suppressing the chlamydial load in koalas. The combined vaccines also induced an increase in anti-KoRV IgG levels, which reduced KoRV-B expression [1]. A recombinant chlamydial MOMP-adjuvanted vaccine was shown to reduce *C. pecorum* load and clinical disease progression in wild koalas [65], suggesting its efficacy in reducing chlamydial burden. Notably, a recent study reported the data of a 10-year assessment of the effectiveness of an MOMP-based vaccine in wild koalas from South East Queensland, where vaccinated koalas exhibited a significantly lower disease incidence, with a 64% reduction in chlamydial infection-related mortality [66]. Overall, MOMP-based chlamydial vaccines show promise for further development and clinical use.

## 7. Conclusions

*C. pecorum* is the most pathogenic and prevalent chlamydial species in koalas. *C. pecorum* is endemic to many koala populations and has significant effects on their health and longevity. However, the current knowledge on *C. pecorum* is limited, with an insufficient understanding of the immune response in koalas, owing to insufficient analytical tools. Further focus should be placed on novel control and prevention strategies, such as the development of new therapeutic interventions and effective vaccines that will benefit koala conservation strategies and improve long-term survival.

## Figures and Tables

**Figure 1 animals-14-02686-f001:**
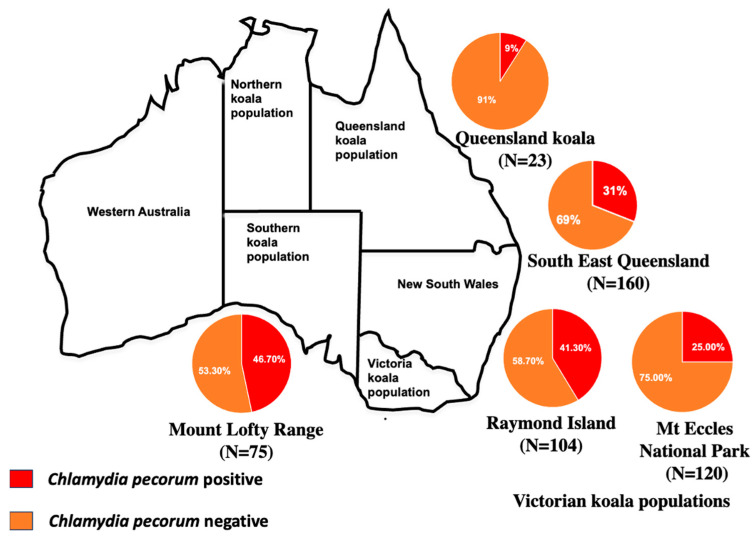
Epidemiology of *Chlamydia pecorum* infection in Australian koala populations.

**Figure 2 animals-14-02686-f002:**
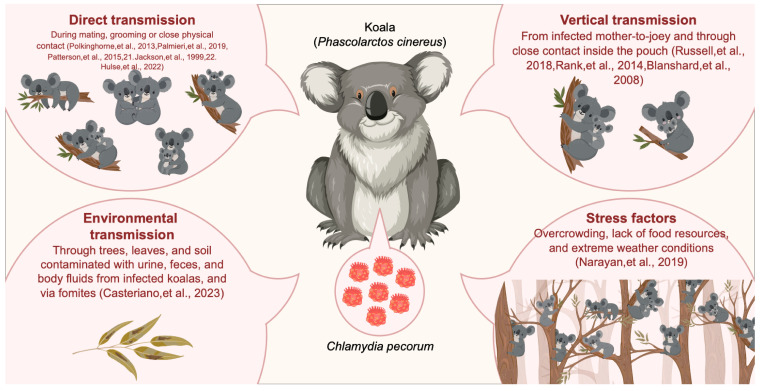
Mode of transmission of Chlamydia in koala.Direct transmission [2,3,11,21,22]. Environmental transmission [24]. Vertical transmission [23,25,26]. Stress factors [27].

**Figure 3 animals-14-02686-f003:**
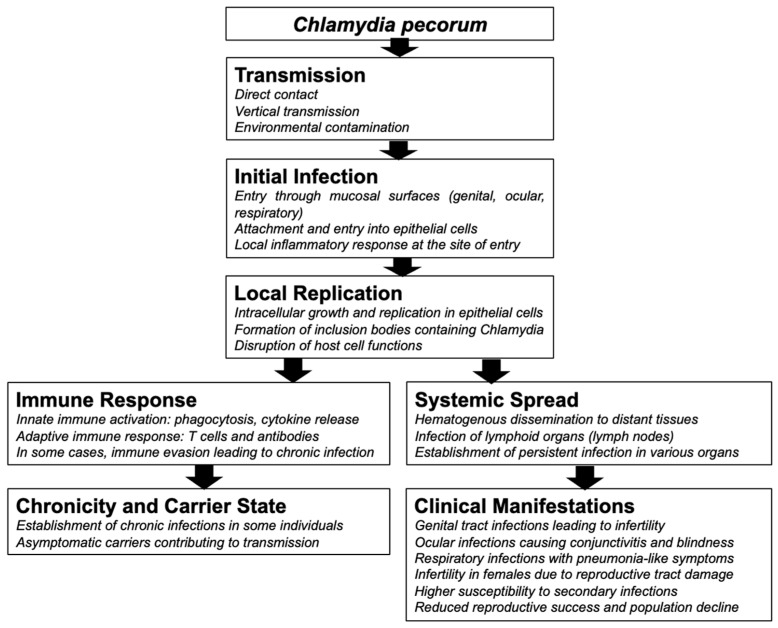
Pathogenesis of chlamydial infection in koalas.

## Data Availability

The data presented in this study are available on request from the corresponding author.

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
