# Peer review of "Epidemiology, Transmission Mode, and Pathogenesis of Chlamydia pecorum Infection in Koalas (Phascolarctos cinereus): An Overview"

_animals, 2024, doi:10.3390/ani14182686_

Round 1
Reviewer 1 Report
Comments and Suggestions for Authors
Overall, this review only briefly presents what is known of chlamydial infection in koalas, and if more comprehensive, would add more significantly to the existing scientific literature as a review. Also, the manuscript lacks flow, with sentences of different topics within paragraphs occurring frequently. The authors need to group the topic within paragraphs to ensure proper organisation of the content of the review. In addition I have detailed specific changes that need to be made in the attached document.

Comments on the Quality of English LanguageThere are some minor issues with grammar in the manuscript.
Author Response
Overall, this review only briefly presents what is known of chlamydial infection in koalas, and if more comprehensive, would add more significantly to the existing scientific literature as a review. Also, the manuscript lacks flow, with sentences of different topics within paragraphs occurring frequently. The authors need to group the topic within paragraphs to ensure proper organization of the content of the review.
Response: We are grateful to the reviewer for careful reading of the manuscript and comments. We have taken every comment into consideration and updated accordingly, therefore we believe that the revised manuscript is in good shape now.
In addition, I have detailed specific changes that need to be made below:.
Line 44: The species identification ‘C psittaci’ is based on a paper in 1988 before the reclassification of koala infections to koala-specific C pecorum and C pneumoniae using molecular techniques. Please delete the inclusion of C. psittaci and the outdated reference 5.
Response: We thank the reviewer for careful reading of the manuscript and suggestions. Accordingly, we have deleted the inclusion of C. psittaci and the outdated reference 5.
Line 46: Studies focus on C pecorum now as it is recognized as the more prevalent and pathogenic species in koalas compared with C. pneumoniae
Response: We thank the reviewer and we have replaced the sentence with the more suitable sentence as suggested by the reviewer (lines 64-65).
Line 47-49 does not flow, reword: Although the role of chlamydiosis in population decline remains poorly understood [2,11]; it is believed to be affecting koala health and koala’s long-term survival [12,13].
Response: We are grateful to the reviewer for careful reading and comments. We have reworded to maintain the flow of readability (line 66).
Line 50: This sentence would fit better at the end of the first paragraph: C. pecorum is globally known as the ‘koala chlamydia’ [14].
Response: According to reviewer suggestion, we have moved this sentence (line 52).
Line 64: Please clarify the meaning of this sentence with rewording: ‘studies impacting proper handling of chlamydial infections [16,17].’ Perhaps handling should be substituted with ‘understanding’?
Response: We thank the reviewer for the comment. We have replaced ‘handling’ with ‘understanding’, as suggested by the reviewer (line 82).
Line 82: The interpretation of ocular and urogenital swab results could also indicate differing pathogenesis in this population (‘suggesting urogenital swab is more sensitive compared to ocular swab for Chlamydia screening’)
Response: We thank the reviewer for the comment. We have added this interpretation (lines 134-135).
Line 84: It would be better to keep the Queensland prevalence comparisons with the previous section where it is first mentioned in lines 69-72.
Response: We thank the reviewer for the comments. Accordingly, we have moved it to the previous section (lines 55-59).
Line 85: ‘Upper and lower reproductive tract’- how would this be defined in male koalas? Usually, these terms are applied to the urinary tract rather than the reproductive tract, and I would suggest removing these inappropriate terms to focus on just presence in both females and males.
Response: We thank the reviewer. In line with reviewer's comments, we have updated it (line 55).
Lines 85-90: This anatomical site information would be better placed at the end of the anatomical site paragraph line 42.
Response: We thank the reviewer. In line with the reviewer's comments, we have accordingly rearranged the sentences (lines 55-59).
Line 94: The sentence beginning ‘C. pecorum 94 gastrointestinal tract (GIT) infection is associated with urogenital tract…’ would also be better placed at the end of the paragraph line 42.
Response: We thank the reviewer. In line with the reviewer's comments, we have accordingly moved the sentences (lines 52-55).
Line 98-99: Delete ‘It was reported that two distinct strains of C. psittaci can infect koalas and should be associated with conjunctivitis and urogenital tract infection [5].’ As this reference is outdated.
Response: As per the reviewer's comments, we have deleted these sentences and references.
Lines 99-101: Move this line to the Queensland paragraph lines 69-72: ‘In free-ranging koalas in southeastern Queensland a higher prevalence (71%, 46/65) of C. psittaci was observed, however, only six (9%) of these koalas showed clinical signs of disease [22].’
Response: In response to reviewer's comment, we have moved these sentences to the suggested paragraph (lines 94-97).
Lines 101-108: Delete section beginning ‘An earlier study also reported the presence of C. psittaci in captive koalas…’ as this is outdated. Delete reference 23
Response: In line with the reviewer's comment, we have deleted these sentences and other sentences of C. Psittaci and the references.
Figure 1: The koala image could be of higher quality.
Response: In response to reviewer's comment, we have improved the koala image (modified figure 2).
Figure 2: the immune response to infection is not referenced
Response: We thank the reviewer for careful reading and pointing out it. In response to reviewer's comment, we have added an immune response to chlamydial infection in a separate subsection (lines 314-345).
Lines 185-191: female reproductive pathology should be with the male in paragraph starting line 161.
Response: We thank the reviewer for careful reading and for the suggestion. Accordingly, we have moved it to the more relevant section. In addition, we also have deleted any repetition (lines 268-273).
Lines 192-195: Likewise, all the reproductive pathology should be together first
Response: As per reviewer comments, we have put reproductive pathology together in the revised manuscript (lines 277-282).
Lines 196-200: Chlamydial load discussion should be elsewhere
Response: As per reviewer comments, we have moved elsewhere (lines 60-62).
Line 202: potential dysbiosis
Response: We have updated (line 348).
Line 204: Although wild koalas infected with Chlamydia are treated with antibiotics in hospitals, however a large percentage of the infected animals show treatment failure and do not survive [37].
Response: We have updated (line 350).
Line 221: Reword for clarity: ‘Scientists are trying to develop an effective vaccine to fight against well spread and debilitating effects of chlamydial infection in koalas’. Substitute widespread for well spread?
Response: We are grateful to the reviewer for very careful reading of the manuscript. We have reworded it for clarification, as suggested by the reviewer (line 425).
Line 227: ’It has been shown that MOMP could be 227 as an antigen for developing chlamydial vaccine [44]’ precedes the same study results using MOMP- this does not make sense, delete the line shown here.
Response: In line with reviewer's comment, we have deleted it.

Reviewer 2 Report
Comments and Suggestions for Authors
Manuscript review
Overall this is a useful review of the issues around chlamydial infections in koalas. Given that it is relatively short, there are sections that are either missing or could be expanded. For example, there should be a semi-detailed section on immune response in the koalas to chlamydial infections. Also, the section on vaccines could be expanded to include perhaps some more recent publications on this topic.
Throughout the manuscript the grammar and sentence structure is quire por. This needs to be improved prior to any publication.
The Epidemiology section is quite interesting. Could a summary Figure perhaps be inserted.
Figure 1 : Some routes of transmission are proven while others are purely speculative. These differences should be made clear.
Figure 2 is very good.
Comments on the Quality of English LanguageSee attached
Author Response
Overall this is a useful review of the issues around chlamydial infections in koalas. Given that it is relatively short, there are sections that are either missing or could be expanded. For example, there should be a semi-detailed section on immune response in the koalas to chlamydial infections. Also, the section on vaccines could be expanded to include perhaps some more recent publications on this topic.
Response: We thank the reviewer for careful reading of the manuscript and for useful suggestions and comments. In line with the reviewer comments, we have added a new section titled ‘immune response to chlamydial infection’. In line with reviewer comments, we also have expanded vaccine section including some recent related publications (lines 315-346; 435-446; 458-461).
Throughout the manuscript the grammar and sentence structure is quiet. This needs to be improved prior to any publication.
Response: We thank the reviewer for the comments. We had our paper edited by Editage.
The Epidemiology section is quite interesting. Could a summary Figure perhaps be inserted.
Response: We thank the reviewer for the appreciation. In response to the reviewer's comment, we have added a summary figure on epidemiology (new figure 1).
Figure 1: Some routes of transmission are proven while others are purely speculative. These differences should be made clear.
Response: We thank the reviewer for the comments. As per reviewer suggestions, we have updated the text (lines 245-247).
Figure 2 is very good.
Response: We are grateful to the reviewer for their sincere comment.

Round 2
Reviewer 1 Report
Comments and Suggestions for Authors
The authors have responded to specific correction suggestions, but not to the overall flow and readability issues in the paper. In every paragraph, the content of the sentences do not flow from one to the other, making it very difficult to read. The sentences appear as lists of unconnected facts under a general paragraph theme, rather than being linked together. The manuscript needs to be properly proof read throughout to ensure that the paragraphs are well presented and cohesive. Also, there is still reference to C. psittaci from reference 21, Figure 2 is still an inappropriate image, and the content of the Figures lack referencing, as previously pointed out. However the main focus should be that the authors need to invest time to correct the issues with sentence flow that begin in the first paragraph and exist throughout the entire manuscript, to ensure that the manuscript quality is at a publishable standard.
Comments on the Quality of English LanguageThis manuscript is not well written in regards to flow and readability, an ongoing issue which has not been addressed in this revision, and therefore is not at a standard for publication at this time.
Author Response
The authors have responded to specific correction suggestions, but not to the overall flow and readability issues in the paper. In every paragraph, the content of the sentences do not flow from one to the other, making it very difficult to read. The sentences appear as lists of unconnected facts under a general paragraph theme, rather than being linked together. The manuscript needs to be properly proof read throughout to ensure that the paragraphs are well presented and cohesive. Also, there is still reference to C. psittaci from reference 21, Figure 2 is still an inappropriate image, and the content of the Figures lack referencing, as previously pointed out. However the main focus should be that the authors need to invest time to correct the issues with sentence flow that begin in the first paragraph and exist throughout the entire manuscript, to ensure that the manuscript quality is at a publishable standard.
Response: Thank you for this comment. As per the reviewer’s comments, to ensure that the paragraphs are well presented and cohesive, we had our manuscript edited using Editage’s English editing service. We have also removed reference 21. We have modified and added citations in figure 2.
English language
This manuscript is not well written in regards to flow and readability, an ongoing issue which has not been addressed in this revision, and therefore is not at a standard for publication at this time.
Response: Thank you for the comment. We have edited our manuscript using the professional editing service Editage (a certificate is attached) to improve the language issues raised by the reviewer.

Reviewer 2 Report
Comments and Suggestions for Authors
Good changes made to the manuscript
Comments on the Quality of English LanguageGood changes made to the manuscript now
Author Response
Thank you very much for your kind review.
